# Controlling COVID-19 via test-trace-quarantine

Cliff C. Kerr [1✉], Dina Mistry[1,8], Robyn M. Stuart [2,3,8], Katherine Rosenfeld [1], Gregory R. Hart[1], Rafael C. Núñez[1], Jamie A. Cohen[1], Prashanth Selvaraj [1], Romesh G. Abeysuriya[3], Michał Jastrzębski[4], Lauren George [1], Brittany Hagedorn[1], Jasmina Panovska-Griffiths [5,6], Meaghan Fagalde[7], Jeffrey Duchin[7], Michael Famulare[1] & Daniel J. Klein[1]

Initial COVID-19 containment in the United States focused on limiting mobility, including school and workplace closures. However, these interventions have had enormous societal and economic costs. Here, we demonstrate the feasibility of an alternative control strategy, test-trace-quarantine: routine testing of primarily symptomatic individuals, tracing and testing their known contacts, and placing their contacts in quarantine. We perform this analysis using Covasim, an open-source agent-based model, which has been calibrated to detailed demographic, mobility, and epidemiological data for the Seattle region from January through June 2020. With current levels of mask use and schools remaining closed, we find that high but achievable levels of testing and tracing are sufficient to maintain epidemic control even under a return to full workplace and community mobility and with low vaccine coverage. The easing of mobility restrictions in June 2020 and subsequent scale-up of testing and tracing programs through September provided real-world validation of our predictions. Although we show that test-trace-quarantine can control the epidemic in both theory and practice, its success is contingent on high testing and tracing rates, high quarantine compliance, relatively short testing and tracing delays, and moderate to high mask use. Thus, in order for test-trace-quarantine to control transmission with a return to high mobility, strong performance in all aspects of the program is required.

[1] Institute for Disease Modeling, Global Health Division, Bill & Melinda Gates Foundation, Seattle, WA, USA. [2] Department of Mathematical Sciences, University of Copenhagen, Copenhagen, Denmark. [3] Burnet Institute, Melbourne, VIC, Australia. [4] GitHub, Inc, San Francisco, CA, USA. [5] Department of Applied Health Research, University College London, London, UK. [6] Wolfson Centre for Mathematical Biology and The Queen's College, Oxford University, Oxford, UK. [7] Public Health – Seattle & King County, Seattle, WA, USA. [8] These authors contributed equally: Dina Mistry, Robyn M. Stuart. ✉email: ckerr@idmod.org

Within 12 months of the world first becoming aware of COVID-19, the total number of diagnosed cases exceeded 80 million[1], with the true number of infections likely much higher. As the pandemic has evolved, so too have global public health responses. Many of the initial efforts to contain the spread focused on border controls, but when these proved insufficient to prevent community transmission, the focus turned to broad non-pharmaceutical interventions (NPI), such as lockdowns and other physical distancing measures. Whilst effective in controlling infection rates[2], such measures have come at enormous cost[3]. Consequently, governments are increasingly relaxing lockdowns in favor of more targeted "test-and-trace" strategies, whereby only those most likely to have COVID-19—for example, those who have symptoms, or who have been in contact with a confirmed case—are required to quarantine. Such strategies have the potential to offer the epidemiological benefits of a large-scale lockdown with smaller economic and societal costs.

Several studies have examined test-and-trace-based containment strategies of COVID-19 in different contexts. Modeling studies have provided evidence that the success of such strategies depends on the proportion of symptomatic cases, the speed and completeness of contact tracing, and adherence to isolation and quarantine[4–8]. Given an estimated basic reproduction number ($R_0$) of 2.4–5.6[9], the number of effective contacts must be reduced by at least 60–80% to achieve epidemic control. Despite this stringent requirement, China successfully demonstrated the feasibility of epidemic control through mandatory home-based quarantine and isolation of those with confirmed infections. This strategy was successfully followed by South Korea, Singapore, and other countries[10]. However, success has not been universal, and some countries have had to re-impose restrictions due to epidemic resurgence after relaxing social distancing restrictions[11,12].

These studies show that there are global precedents for the success of containment strategies based on (a) high rates of routine testing (including both symptomatic and asymptomatic testing), (b) rapid return of test results, (c) high rates of contact tracing, and (d) social support for people who have been diagnosed or quarantined—a strategy we refer to as test-trace-quarantine (TTQ; also known as test-trace-isolate). However, the success of this strategy depends on how effectively each component is implemented. To date, the COVID-19 response in the United States (and other Western countries, such as the United Kingdom) has been marked by insufficient quantities of test kits and associated supplies, along with challenges in implementing contact tracing at scale, and imperfect adherence to isolation guidelines[13]. Increasing coverage of infection-blocking vaccines will make control easier[14], but this effect is likely to be at least partially countered by the higher transmissibility, and potential immune escape, of emerging SARS-CoV-2 variants[15].

This study investigates what the requirements are for a high-income, urbanized setting to successfully transition from a policy of mobility restrictions towards TTQ-based containment prior to high vaccine coverage. To answer this question, we use Covasim, a detailed, data-driven, agent-based model of COVID-19, and apply it to the Seattle context (specifically King County, which includes Seattle and the surrounding metropolitan area).

## Results

### Mobility restrictions achieved initial epidemic control.
The first case of SARS-CoV-2 in the USA was diagnosed on 20 January 2020 in the Seattle metropolitan area[16]; the first documented COVID-19 death in the USA was on 27 February, a resident of a long-term care facility (LTCF) also in the Seattle area[17]. Local and state governments began issuing a series of measures to control the expanding epidemic, including school closures on 12 March and a shelter-in-place order (Stay Home, Stay Healthy) from 23 March until 31 May 2020[18].

We fit the Covasim model to age-stratified data on COVID-19 diagnosed cases and deaths in Seattle from January through June 2020 using Optuna, a parameter optimization library[19]. Data inputs included detailed demographic information (including population age structure and contact patterns, school enrolment, employment, and LTCF residency), mobility data (provided by SafeGraph; see http://safegraph.com), and COVID-19 testing data. As shown in Fig. 1, Covasim was able to accurately reproduce the detailed time trends of both diagnoses and deaths (Fig. 1a, b), including the age distribution of each (insets). We estimate that approximately 100,000 SARS-CoV-2 infections (95% confidence interval: 80,000–115,000 infections) occurred in Seattle between 27 January and 9 June 2020 (Fig. 1c), out of a total population of 2.25 million, for an attack rate (cumulative infections divided by population size) of 3.5–5.1%. A total of 8548 cases had been diagnosed by June 9, for an overall diagnosis rate of 9% (95% CI: 7–11%). The effective reproduction number, $R_e$, is estimated to have been 2.3 (95% CI: 2.0–2.6) prior to policy interventions, consistent with previous estimates[20], and to have dropped below 1 as the shelter-in-place order took effect (Fig. 1d). This period also coincided with the peak number of active infections, 16,000, with model projections validated by prevalence data from the Seattle Coronavirus Assessment Network (Fig. 1c).

Calibrated model parameters, which provide estimates of transmission dynamics and intervention effectiveness, are shown in Fig. 1e. The parameters used for calibration were: overall transmissibility $\beta$, defined as the probability of transmission between an infectious and susceptible adult on a single day in a typical household setting; transmission rates relative to baseline, which may change due to mask usage, hygiene, physical distancing, and other measures; and the odds ratio for people with COVID-19 symptoms being tested vs. people without symptoms (i.e., uninfected, asymptomatic, or presymptomatic people). To determine the impact of mobility-related changes in transmission, we calibrated the model using reductions in the number of work and community contacts based on SafeGraph weekly mobility data (M, blue), as well as using no mobility data (N, orange). Including the mobility trends, we found that relative transmissibility was reduced by 12 ± 5% compared to its initial value, reflecting the impact of other NPI, including interpersonal distancing, hygiene, and mask use; this drop is consistent with increasing trends of protective health behaviors[21]. To verify the calibration, we excluded mobility data and recalibrated the model, finding that relative transmissibility dropped by 71 ± 3% compared to its initial value. All other parameters had consistent values between the two calibrations, including the change in transmission at LTCFs (estimated to have dropped by 80–92%) and overall transmissibility (estimated at 4.3–4.5% per household contact per day). The symptomatic testing odds ratio, reflecting the much higher rate at which people with COVID-like symptoms test, was estimated to be 17–24. While the testing odds ratio remained constant, the routine testing yield (the number of diagnoses divided by the number of tests) showed a decline from 10–15% in March to 1.5–2.5% in early June, due to the much lower number of active infections.

Since Covasim includes intra-host viral dynamics and a detailed demographic model, it can be used to investigate mechanisms of transmission, as shown in Fig. 2. We found that infections were primarily driven by transmission in workplace and community contact layers (accounting for approximately 58% of the total) prior to interventions. Surprisingly, even though distancing interventions led to a roughly two-thirds drop in workplace and community mobility (Fig. 1f), the total proportion

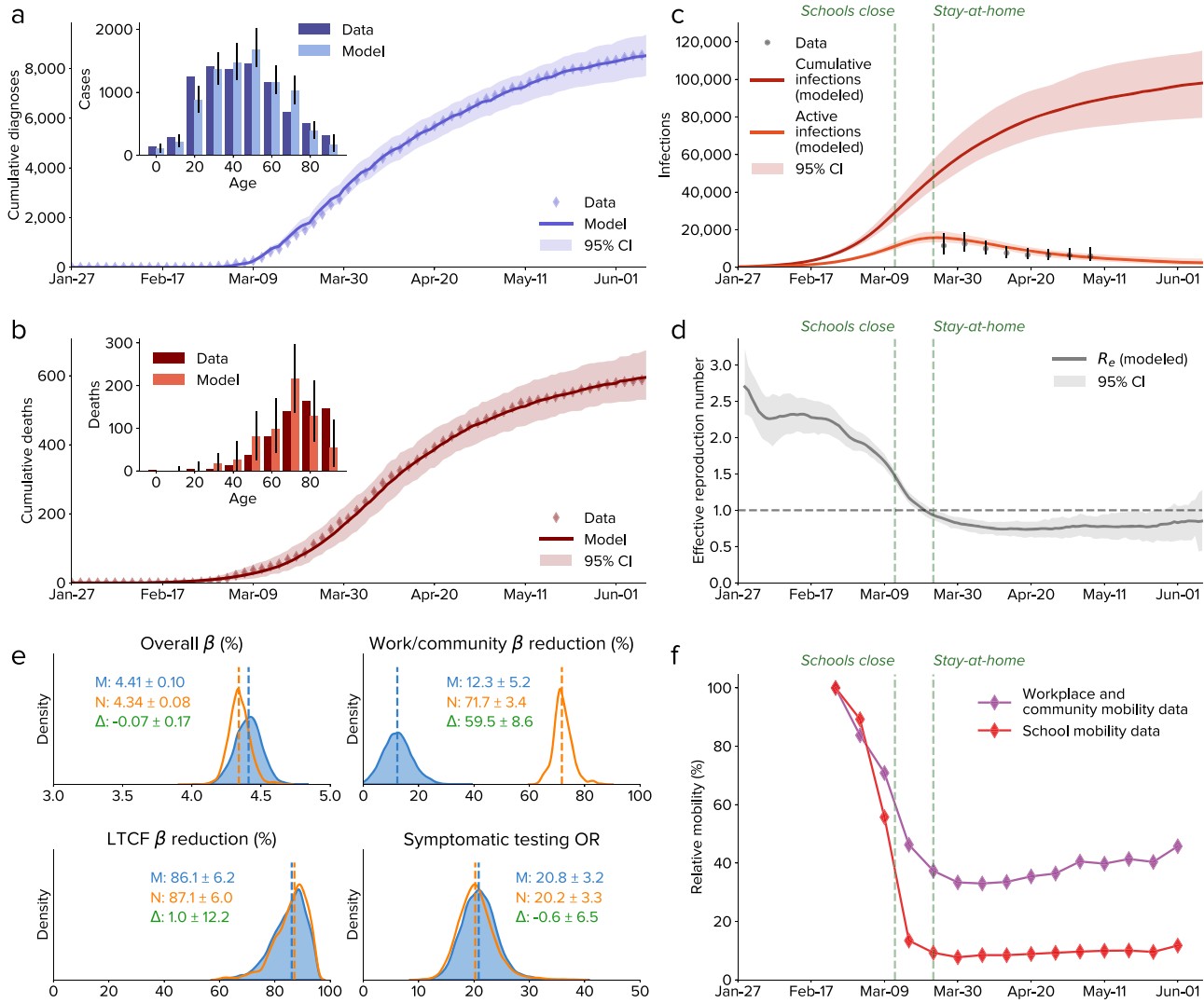

**Fig. 1 Calibration of the model to data from Seattle-King County, Washington, from 27 January to 9 June 2020. a-b** The cumulative number of diagnosed cases and deaths, over time and by age. **c** Estimated numbers of cumulative and active infections. Dashed lines show policy interventions; data are from the Seattle Coronavirus Assessment Network. **d** Effective reproduction number, showing a drop consistent with policy interventions. **e** Parameter distributions for 23,913 simulations calibrated either with SafeGraph mobility data (M, blue) or with no mobility data (N, orange); the difference (Δ, green) is only significant for work/community transmission reduction. **f** SafeGraph mobility data for workplaces and the community and for schools. CI, confidence interval; β, transmission rate; LTCF, long-term care facility; OR, odds ratio. In all panels, values show medians, and ranges show 95% confidence intervals.

of infections due to workplace and community transmission reduced only slightly, to 52% (Fig. 2a). This is in part due to the high overdispersion of SARS-CoV-2 infections (Fig. 2b): a majority of people infected do not transmit, while 50% of infections are caused by just 10% of people infected; these 10% infect, on average, 6.3 other people (Fig. 2c). Thus, a relatively small proportion of highly infectious individuals are likely responsible for a majority of ongoing COVID-19 spread. Preliminary data from the contact tracing program in Seattle provides further evidence for this: of the 44% of household contacts who received a COVID-19 test, 43% of them tested positive (i.e., 19% of traced household contacts were positive); of the 31% of non-household contacts who were tested, 28% tested positive (i.e., 9% of traced workplace contacts were positive). High-risk index cases and contacts were preferentially both traced and tested, so these estimates represent an upper bound on the attack rate, and international estimates on household secondary attack rate have been even lower, ranging from 5%[22] to 19%[23]. To be consistent with our estimated value of $R_e$, these relatively low household attack rates require high dispersion and significant

non-household transmission. We also find that 54% of transmissions are from symptomatic individuals, similar to previous estimates[24,25].

**Idealized test-trace-quarantine results in self-limiting epidemic dynamics.** Before investigating TTQ in the Seattle context, we first consider how TTQ impacts SARS-CoV-2 transmission in a hypothetical population. Consider an idealized TTQ scenario, where all contacts are traced, all traced contacts are tested and enter into 14-day quarantine (regardless of test result), and combined testing and tracing delays are less than the duration of infectiousness (which is also assumed to be the time period when a person would test positive). In this idealized scenario, epidemic control can be achieved even for high values of $R_0$, regardless of the stage of the epidemic at which the intervention begins. This is because as a branch from a cluster of infections grows, the probability increases that someone from that branch will be diagnosed. When this occurs, idealized contact tracing would identify that branch via a series of steps, including both backward

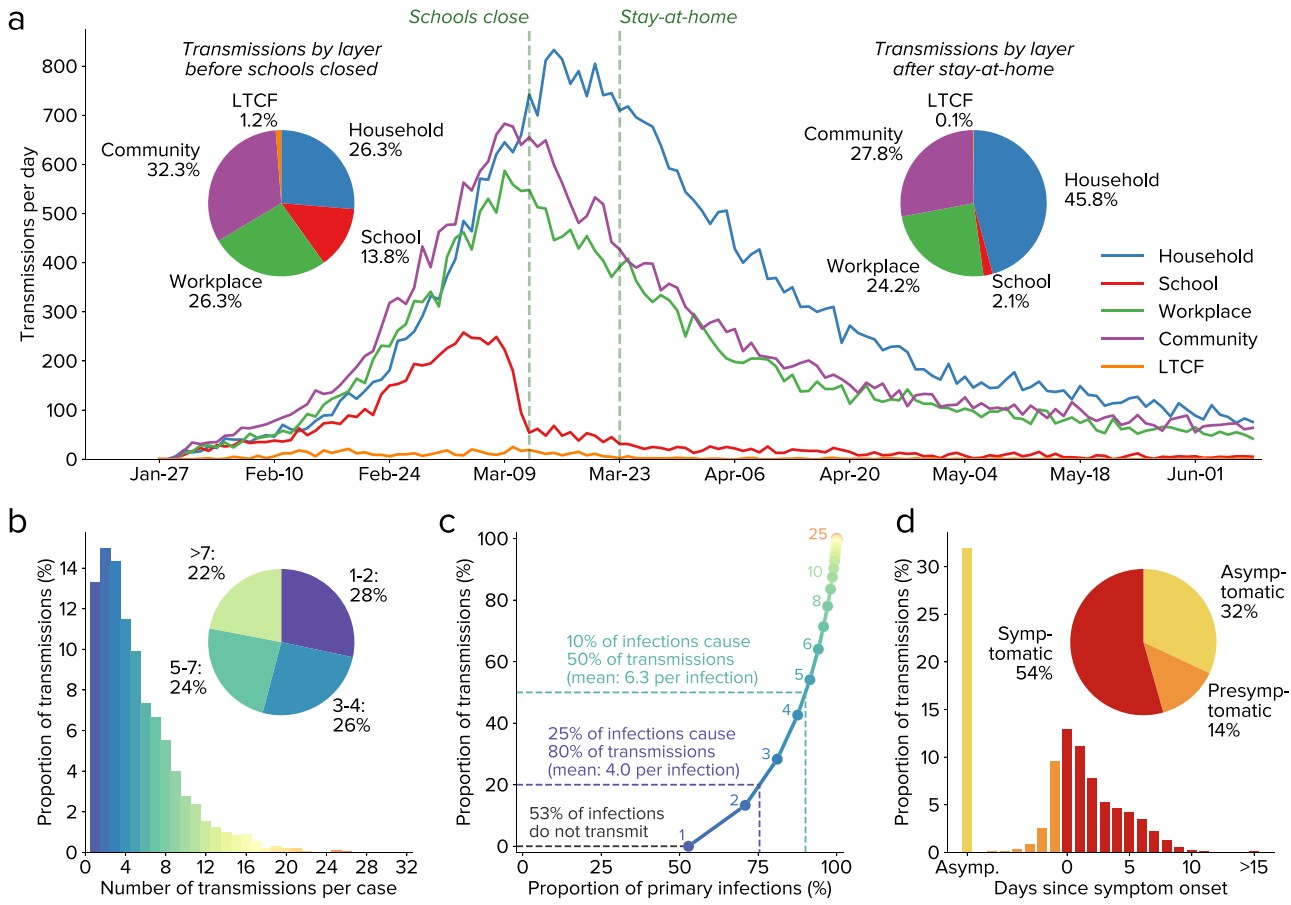

**Fig. 2 Modeled transmission dynamics. a** Infections over time by contact layer. **b** Overdispersion of infections (up until school closures on 12 March), with roughly equal numbers of infections attributable to individuals who transmit to 1–2 others, 3–4 others, 5–7 others, or more than 7 others. **c** Due to overdispersion, 53% of all primary infections do not cause any secondary infections, while 10% of primary infections are responsible for 50% of secondary infections. Annotations show the number of transmissions per primary infection, corresponding to each bar of panel **b**. **d** Infections as a function of symptom onset, showing that slightly over half of infections are transmitted by symptomatic individuals.

(upstream) and forward (downstream) infections[26], hence removing that branch from the infectious pool.

Since each traced contact who tests positive results in additional traced contacts, contact tracing can be thought of as an "infectious" process on the network[27]. Specifically, if (a) the sum of the testing and tracing delays is less than the average serial interval of SARS-CoV-2; and if (b) the majority of secondary transmissions are successfully traced, diagnosed, and isolated, then the number of traced and diagnosed contacts will spread locally on the network faster than SARS-CoV-2 infections, extinguishing that branch. The theoretical maximum number of backwards steps that can be taken is approximately the duration for which someone returns a positive test following infection divided by the sum of testing and tracing delays. Assuming the former is approximately 10–14 days and the latter is approximately 2–4 days, roughly 2–5 backward steps are theoretically possible. In practice, false negative tests would likely break the chain sooner, although Japan, Vietnam, and Australia have successfully backwards-traced contacts for up to 14 days, in some cases by tracing secondary contacts before test results are returned[28]. However, even with just forward tracing, epidemic control is still theoretically achievable. Several recent studies have produced similar findings[26,29,30].

Figure 3 shows an illustrative example of idealized TTQ resulting in epidemic control. In a hypothetical population of 100 people without interventions, infections continue until herd immunity prevents further spread (Fig. 3a). If a high level of

testing and isolation is introduced (15% probability of testing per day for people with symptoms, coupled with 80% effective isolation), the number of infections is only modestly decreased despite roughly half of cases being diagnosed, since a large proportion of transmission occurs before cases are diagnosed. Adding a moderate level of tracing (70% of household contacts, 10% of workplace and school contacts) significantly reduces the number of infections (Fig. 3c), due to the rapid diagnosis of traced contacts and the preventative effect of quarantine.

Crucially, we find that the effectiveness of contact tracing is proportional to incidence, thereby resulting in self-limiting epidemic dynamics. Figure 3d shows a hypothetical population of 30,000 people in a medium transmission scenario ($R_0 = 2.5$), where borderline epidemic control ($R_e \approx 1$) can be achieved through either moderate physical distancing alone (i.e., 60% reduction in $\beta$), high levels of routine testing and isolation alone (75% daily probability of people with symptoms testing and isolating), or TTQ (8% daily probability of people with symptoms testing, 90% of contacts of diagnosed individuals being traced and quarantined, and 75% probability of testing on entering quarantine). In a low transmission setting (Fig. 3e, $R_0 = 2.0$), both physical distancing and testing lead to rapid epidemic extinction, while TTQ maintains $R_e \approx 1$. Conversely, in a high transmission setting (Fig. 3f, $R_0 = 3.1$), TTQ again maintains $R_e \approx 1$, while physical distancing and testing do not achieve epidemic control. This is because distancing and routine testing act like constant multipliers on transmission; they will achieve epidemic

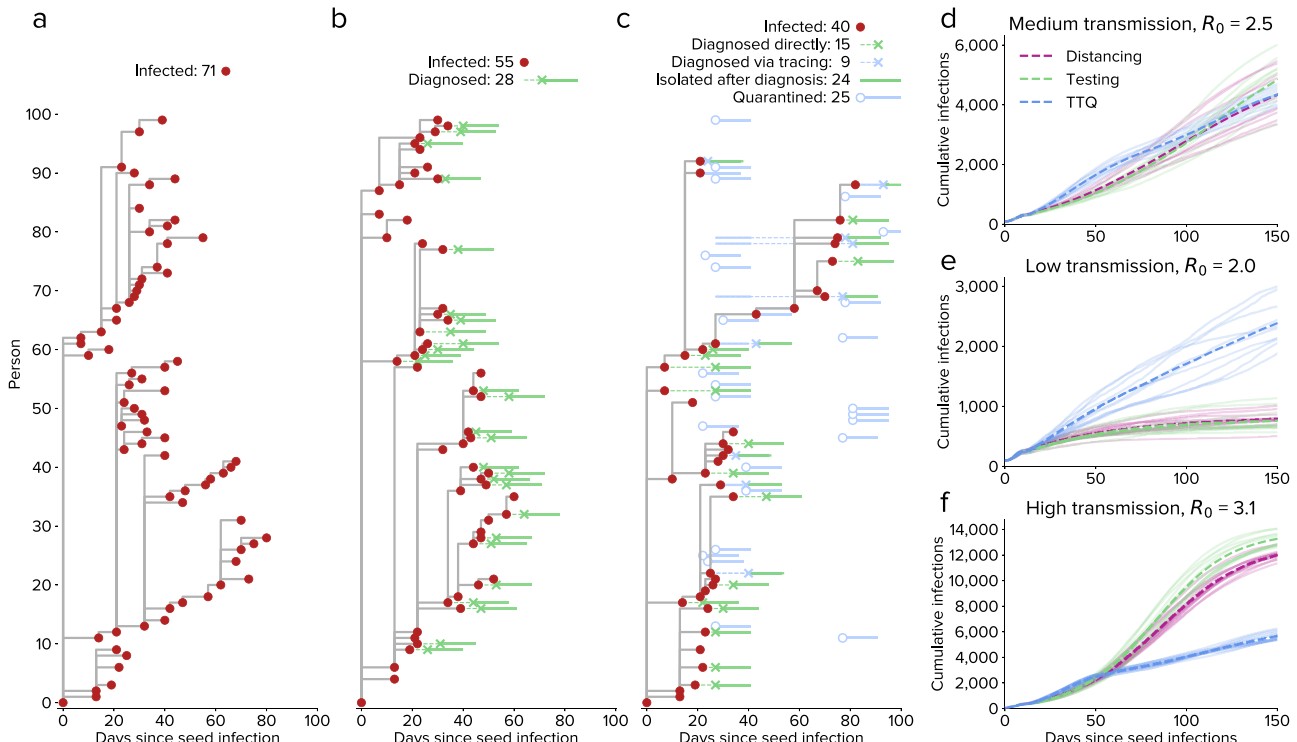

**Fig. 3 Epidemic dynamics differ depending on the intervention. a–c:** Transmission trees for a cluster of 100 people under three scenarios: (**a**) no interventions, (**b**) testing and isolation only (starting on day 20), and (**c**) test-trace-quarantine. **d–f** Comparison of different levels of baseline transmissibility for social distancing only, testing only, or test-trace-quarantine (TTQ). For medium baseline transmission (**d**), moderate distancing, high testing, or high tracing each result in $R_e \approx 1$. For low transmission (**e**), the same distancing and testing interventions both result in $R_e < 1$, while the same tracing intervention maintains $R_e \approx 1$. For high transmission (**f**), the same distancing and testing interventions both result in $R_e > 1$, while the same tracing intervention continues to maintain $R_e \approx 1$.

control if and only if they bring $R_e < 1$. In contrast, in a TTQ scenario with no capacity constraints, more infections will result in more diagnoses, more contacts being traced, more people placed in quarantine, and more people tested in quarantine. This mechanism limits the size of individual clusters of infections in a TTQ setting, as well as placing an upper bound on epidemic growth even with extremely high transmission scenarios (e.g., $R_0 = 5$). However, this phenomenon only occurs with sufficient levels of contact tracing: analogous to $R_e$, self-limiting dynamics only occur if at least one new case is detected on average for each new index case whose contacts are traced. Whether or not this occurs is determined by the probability of contact tracing, the probability of testing in quarantine, the population network structure, and the rate of transmission.

**Realistic test-trace-quarantine scenarios allow high mobility.** Implementing a successful TTQ strategy requires solving a challenging prioritization problem: whom to test, whom to trace, how to ensure people safely isolate and quarantine, how to quickly return test results, and how to quickly trace contacts. Barriers to high performance in each of these areas are varied, including limited budgets, shortages of equipment and staff, behavioral compliance, and racial and economic inequalities.

Figure 4a shows how six aspects of the TTQ strategy affect the estimated numbers of infections in Seattle: (1) effectiveness of isolation and quarantine (i.e., relative reduction in transmission during the 14-day isolation/quarantine period), (2) the number of routine tests per 1000 people per day, (3) the probability of a person's household and workplace contacts being traced following diagnosis, (4) the proportion of contacts who are tested after they are traced, (5) the delay between when a person is tested and

when they receive their test result, and (6) the delay between when a person receives a positive test result and when their contacts are traced. Using the model calibrated to the Seattle region up until 1 June 2020 and projecting forward for a 90-day period, we consider a hypothetical baseline scenario of high mobility (100%), high testing (~6000 routine tests per day, or 2.7 per 1000 people per day, compared to ~1800 routine tests per day as of 1 June 2020 and ~3500 routine tests per day as of 15 July 2020), and high tracing (70% of all household and workplace contacts traced within 2 days, compared to roughly 30% of household contacts and close to 0% of workplace contacts as of 1 June 2020). This scenario was chosen as the most realistic means of achieving an $R_e$ value of below 1 given a return to 100% mobility (Fig. 4b, green diamond). We then vary individual aspects of the response relative to this baseline.

All six aspects of the TTQ strategy had a significant impact on epidemic outcomes. The most important aspects, in terms of their impact on overall attack rate, were isolation/quarantine effectiveness and routine testing probability. Each diagnosed or quarantined person who fully isolates is estimated to avert 1.2 subsequent infections over the 90-day period, while each routine test conducted (including negative tests) is estimated to avert 0.2 infections. However, speed also matters: an additional one-day delay in returning test results is estimated to result in roughly 4 additional infections per person who tests positive, while a one-day delay in tracing contacts is estimated to result in roughly 2 additional infections for every index case whose contacts are traced. While quarantine testing had the smallest overall impact on the attack rate, an additional 0.4 infections are still estimated to be averted for every index case whose contacts are tested. This is because quarantine testing is highly efficient at

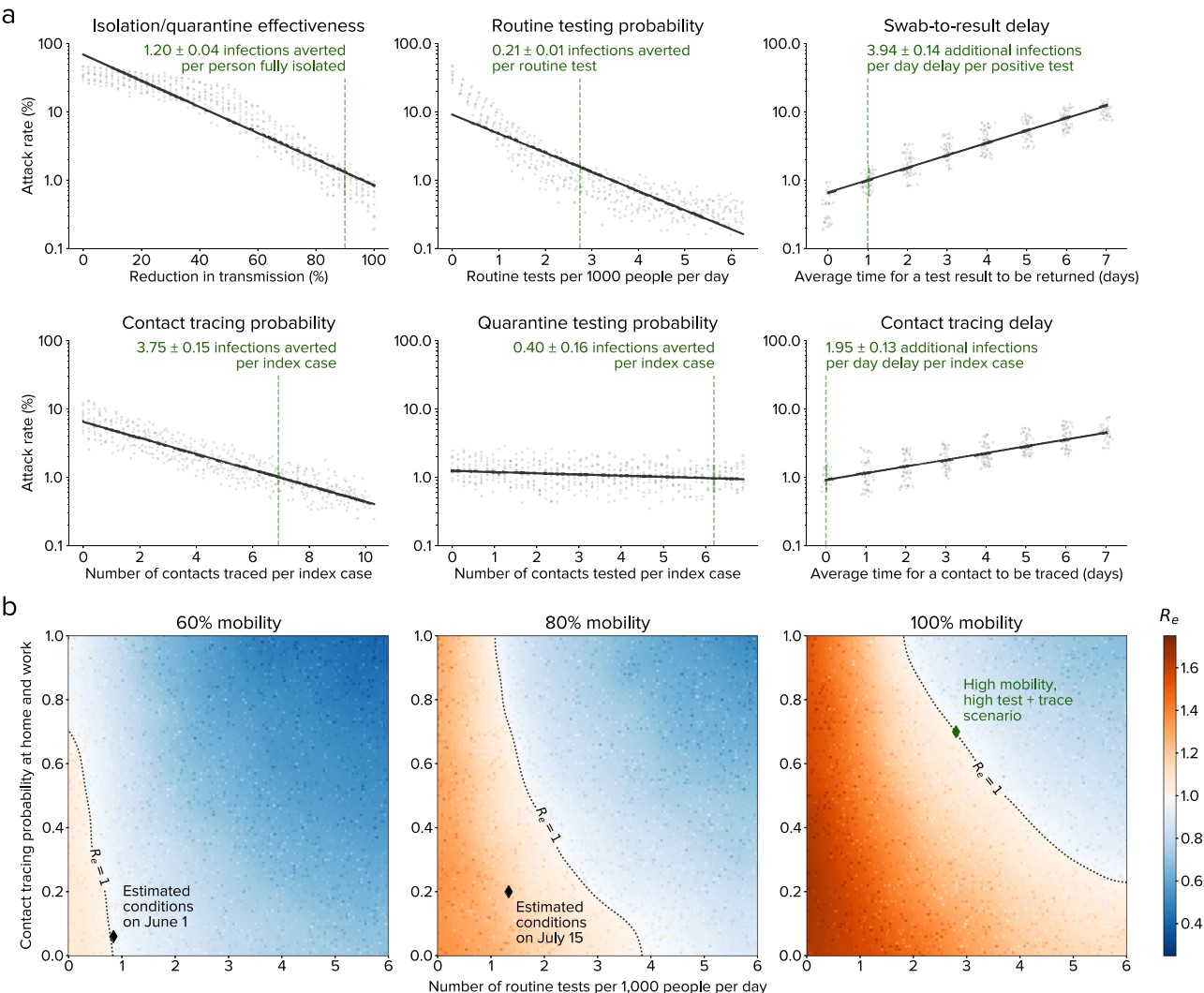

**Fig. 4 Impact of testing, tracing, and quarantine. a** Relative importance of different aspects of the TTQ strategy for a scenario of high mobility (full return to baseline workplace and community movement patterns), high testing, and high tracing in Seattle. Each dot shows a simulation, with other parameters held constant (at the values indicated by the dashed green lines). Low levels of isolation/quarantine effectiveness or routine testing probability lead to the highest attack rates, although all parameters have a significant impact on epidemic outcomes. **b** Countering the effects of increased mobility via testing, tracing, and quarantine. Current interventions (black diamonds) were estimated to keep $R_e$ < 1 for 60% of baseline mobility level (left). Subsequently, increased transmission rates exceeded intervention scale-up, temporarily leading to $R_e$ > 1 (center). For a return to full mobility (right), high levels of both testing and tracing are required to maintain epidemic control (green diamond, corresponding to the dashed lines in panel **a**). Dots show individual simulations.

identifying infections: while the test positivity rate for routine testing in early June was 1.5%, traced contacts had a test positivity rate of 34%.

To explore the practical implications of these results for Seattle, we simulated three reopening scenarios (Fig. 4b): 60%, 80%, and 100% mobility in the workplace and community contact layers, relative to baseline, across a range of testing and contact tracing rates. We assume these mobility changes occur in the context of continued use of masks and other non-mobility-related NPI, which we estimate have together reduced transmission by roughly 10–15% (see Fig. 1e), which is consistent with roughly 40% efficacy[31] coupled with 25–40% compliance. Unsurprisingly, a return to 100% workplace and community mobility in the absence of other interventions would increase $R_e$ to well above 1, leading to a large second wave of infections. High testing and high tracing are both required to maintain epidemic control with full reopening. For example, even a four-fold increase in testing rates would fail to control the epidemic without an increasing in contact tracing.

After the stay-at-home order was lifted on 1 June 2020, there was a large increase in transmission rates, especially among younger age groups, which was followed by a large scale-up of both testing and tracing programs. Figure 5 shows that the model calibrated until 1 June 2020 was able to replicate observed epidemic trends over the next 3 months, providing validation that it is capturing the underlying transmission dynamics in Seattle, including the impact of the testing and tracing interventions on the epidemic. Figure 5 also illustrates what would have been achieved had a "high testing and tracing" scenario, as described above, been implemented instead: while the peak numbers of people tested (Fig. 5a) and contacts traced (Fig. 5b) would have been much higher (5000 vs. 3500 tests per day and 900 vs. 200 contacts traced per day), active infections would have declined much more rapidly (Fig. 5c). By Aug. 31, due to the low number of new infections, we estimate that the number of contacts traced (Fig. 5b) and people diagnosed (Fig. 5d) in the "high testing and tracing" scenario would have been comparable to or lower than the true number.

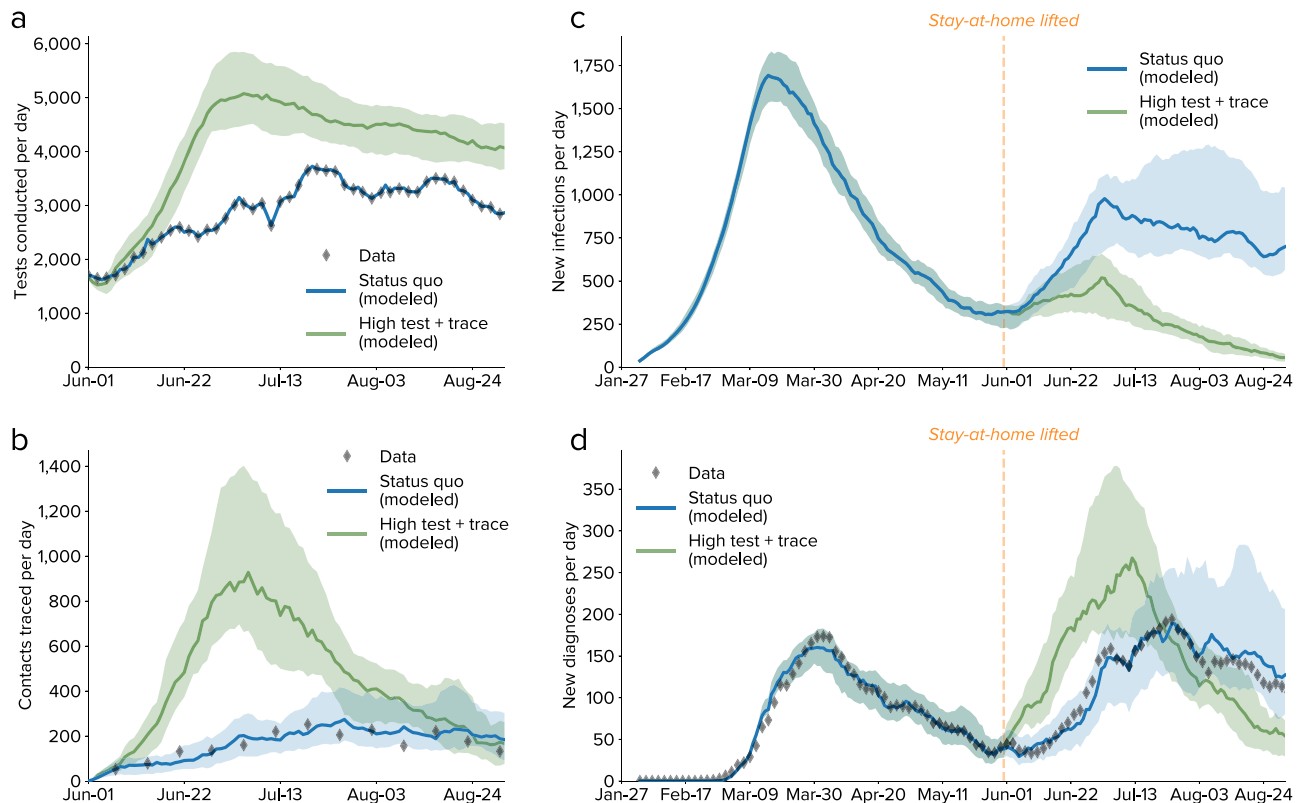

**Fig. 5 Comparison between observed epidemic trends and projected scenarios from 1 June to 31 August 2020. a** Number of tests conducted per day, with modeled values for the status quo (using the data as an input) and a counterfactual scenario with high testing and high tracing. **b** Number of contacts traced per day. **c** Estimated number of new infections, with a significant rise in infections observed shortly after the stay-at-home order was lifted. **d** Number of diagnoses per day, showing consistency between the model and the data both for the calibrated period (27 January–31 May 2020) and the projected period (1 June–31 August 2020). In all panels, lines show medians, and shaded regions show 80% confidence intervals.

## Discussion

Seattle achieved epidemic control between mid March and early June 2020 as a result of (a) greatly reduced mobility; (b) adoption of additional NPI, including interpersonal distancing, hand washing, and face masks; and (c) moderate rates of routine testing, plus low but increasing rates of contact tracing. We separated the effects of the first two components using detailed mobility data, and were therefore able to assess the impact of reopening whilst assuming that distancing, hand washing, and face mask use will continue. We found that with mobility at 60% of its pre-COVID levels, epidemic control could be attained despite relatively low testing and tracing. Returning to full mobility would require identifying and isolating many more cases, by significantly increasing both the number of routine tests conducted and the proportion of contacts traced. From June through September 2020, both mobility and interventions increased, and data from this period were used to validate these findings.

The handful of countries that achieved control of their COVID-19 epidemics without relying on extensive lockdowns have done so using a diversity of approaches, but those that have used a broader set of interventions have achieved more stable epidemic control. Taiwan and South Korea used high rates of testing, contact tracing, mask compliance, and other interventions to quickly bring their epidemics under control[32]. Japan has had high mask compliance and a relatively high rate of contact tracing, but relatively low testing rates; after early control, reported new cases increased from late May through early July 2020[33]. Australia and New Zealand achieved early epidemic control via strong travel restrictions and high rates of testing and contact

tracing. However, mask use was low, and a rapid increase in cases in the state of Victoria, Australia, began in late June 2020, and was only brought under control after reimposed mobility restrictions, mandated mask use, and high testing rates. In contrast, the neighboring state of New South Wales was able to maintain epidemic control without reimposing lockdowns by combining high testing rates (2.9 tests per 1000 people per day, similar to the high testing and tracing scenario presented here) with near-perfect tracing of close contacts[34]. This observational evidence for a diverse range of interventions being required for epidemic control is consistent with our finding that each aspect of the response is roughly equally important: shortfalls in one area (e.g., low rates of testing or mask usage) may be partly offset by high performance in another (e.g., high rates of contact tracing). However, these examples suggest that epidemic control may be only fleeting unless performance is high in all three areas (testing, tracing, and either mask usage or mobility restrictions).

As shown in Fig. 3, control via TTQ produces strong positive feedback dynamics. This means that it is an especially effective control strategy when case numbers are low; however, even relatively small spikes in cases can cause the system to be overwhelmed, a result that has been seen both in theory[35] and in practice[11,12]. If this occurs, without high vaccination rates, then a return to mobility restrictions remains the only consistently effective strategy for regaining control.

There are several limitations of this study. First, we do not consider geographical clustering or day-of-the-week changes in mobility, so cannot model hotspots or outbreaks in specific areas or on specific days. Although this may not be crucial given that interventions were set at a county-wide level, subsequent phases

of the response may require more localized policy actions. Second, there is continued debate around how susceptibility and transmissibility vary by age and with comorbidities; the model parameters reflect the best available evidence to date[36], but new evidence is continually coming to light. Third, there is considerable uncertainty around other crucial characteristics of both SARS-CoV-2 transmission (including the extent to which it is seasonal, and the proportion of asymptomatic and presymptomatic transmission) and the impact of interventions (such as mask efficacy). We have handled these uncertainties by calibrating extensively to data (Fig. 1a–c), and by propagating remaining uncertainties in parameters (Fig. 1e) through all scenarios. However, additional data on transmission characteristics would nonetheless help refine our understanding of the most important transmission pathways, as well as how to disrupt them.

In summary, we have shown that (a) agent-based models can be fit to detailed epidemic time trends and age distributions (Fig. 1), as well as make accurate forecasts (Fig. 5); (b) an idealized test-trace-quarantine program with no capacity constraints can control an epidemic even at high rates of transmission (Fig. 3); and (c) high rates of testing and tracing, short delays, and high quarantine compliance are all important for maintaining epidemic control (Fig. 4a), but the levels required for each are likely to be achievable even under a return to full mobility (Fig. 4b). Thus, we believe the example of Seattle provides strong evidence for test-trace-quarantine as a feasible control strategy—a strategy that other jurisdictions may wish to invest in more heavily.

## Methods

**Model**. We performed modeling using Covasim, an agent-based model of COVID-19 transmission and interventions. Covasim's methodology is described in detail in Kerr et al.[37] Briefly, the Covasim model is designed to capture the nuances of realistic COVID-19 transmission, such as: age and population structure, including relative susceptibility and mortality rates; transmission networks in different social layers, including households, schools, workplaces, communities, and long-term care facilities; and intra-host viral dynamics, reflecting variable infectivity within and between hosts. Covasim also supports an extensive set of interventions: non-pharmaceutical interventions, such as distancing and masks; testing interventions, such as symptomatic and asymptomatic testing, contact tracing, isolation, and quarantine; and pharmaceutical interventions, such as therapeutics and vaccines. While Covasim was originally developed to inform policy decisions in Washington State, it has since been adapted for use in more than a dozen countries, including Vietnam[38], Australia[39], and the United Kingdom[40]. The code is fully open-source (available from GitHub via https://covasim.org); it includes extensive documentation (available at http://docs.covasim.org), tutorials, and software tests. Analyses were performed with Covasim version 2.0.0 and Python 3.8. The following sections describe the customizations to Covasim that were used in this study[41].

**Data and sources**. As an agent-based model, Covasim can make use of rich data sources. Mechanistic representations of individuals, contacts, and infections enables physical parameter values to be input directly or used as priors during model fitting. Default values for most inputs come from publicly available data and literature, and are described in Kerr et al.[37].

To model the Seattle-King County region for this analysis, we used epidemiological data provided by the Washington State Department of Health (WA-DoH) under a use agreement. (Note that "Seattle" and "King County" are used interchangeably, i.e., the analysis is not restricted to the city of Seattle, nor does it include counties in the greater Seattle metropolitan area such as Snohomish and Pierce counties.) WA-DoH maintains all COVID-19 data as a line-list in the Washington Disease Reporting System (WDRS), and has provided weekly exports to the study team for the purpose of conducting this and other analyses in support of model-based decision making. We aggregated line-list entries to daily totals by 10-year age bins to produce target data for model calibration. The resulting dataset includes the number of positive and negative tests (by date of sample collection), as well as the number of deaths (by date of death) in King County. The WDRS records also enabled us to characterize the distribution of delays between symptom onset on diagnostic swab; we used these data to validate the implementation of the testing intervention (Fig. 6).

Using data on the number of tests and number of diagnoses, we were able to calibrate the model to testing yield. By combining this with estimated numbers of infections, which we know from both data on deaths, as well as independent

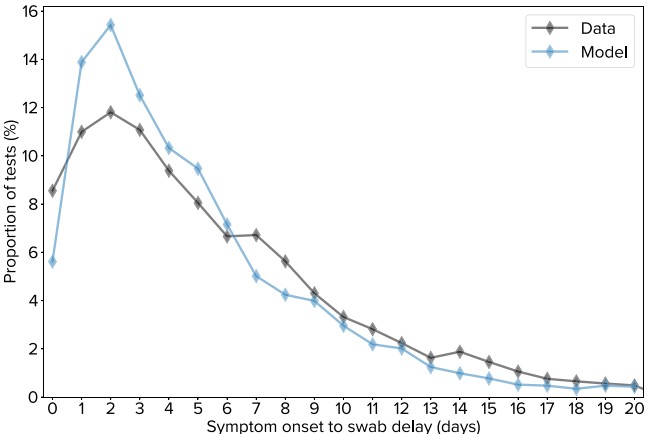

**Fig. 6 Delay from COVID-19 symptom onset to nasal swab sample collection.** Comparison of empirical (black) and simulated (blue) distributions of delays.

seroprevalence surveys (as shown in Fig. 1c), we were able to estimate the testing rates of people both with and without COVID-19. According to the most recent data available at the time the analyses were performed (9 June 2020), roughly 1800 tests per day were being conducted in King County; using this method, we found that a roughly 8% probability per day of testing for people experiencing symptoms, and 0.1% probability per day of testing for people without symptoms (uninfected, asymptomatic, and presymptomatic), allowed us to match observed values for (a) the overall number of tests, (b) the test positivity rate, and (c) the symptom-to-swab delay.

The partnership with Public Health Seattle & King County (PHSKC) has provided additional context to ensure the model captures transmission, testing, care, and contact tracing in this setting. PHSKC co-authors and other county officials have provided data, under use agreement, and insights on testing campaigns, focal outbreaks, schools, hospitalization, and congregate settings such as long-term care facilities (LTCF). Most of these data are available publicly on daily summary and LTCF dashboards. Washington is a home-rule state, meaning that laws can be set at the local level, and as such the contact tracing programs are led by each local health jurisdiction. Seattle-King County was one of the first jurisdictions in the state to pilot contact tracing using local health resources. As of September 2020, the county has sufficient human resources to trace approximately half of the cases, while the other half are handled by the Washington State contact tracing program. Data for this analysis were provided by PHSKC.

We used data on weekly foot traffic patterns obtained through SafeGraph to model the degree of mobility in the workplace and community layers of the model. This publicly available dataset is based on anonymized cell phone data, which connects foot traffic counts with points-of-interest visits for over 5 million unique locations in the United States. This rich dataset enables a detailed view of hourly visits to specific locations. Using an aggregation of visits in King County across all industries starting at the end of January, we classified visits with a dwell-time less than four hours as community-associated mobility, and those visits with a dwell-time of more than four hours as workplace-associated mobility. Using the last week of January as a baseline value for "pre-COVID mobility", we assigned a weekly mobility level for community and work relative to the baseline value.

**Population and network model**. To model King County with detailed information on the demographics and network structures of the King County population, we used SynthPops, an open-source data-driven model for generating realistic synthetic contact networks for populations. SynthPops is available from https://synthpops.org; further details are provided in Kerr et al.[37] For the population of King County, we used a combination of data sources at the county, state, and country resolution with the SynthPops model (version 0.7.2). Specifically, data from the 2018 American Community Survey (ACS) at the county resolution[42] were used to estimate age and household size distributions. The US Census Bureau[43] provided data for the age of reference individuals by household size at the country resolution, and age mixing contact matrices for the US are drawn from Prem et al.[44] For the network layer of schools, we used 2018 ACS 1-year estimates for county enrollment rates by age[42], municipal records on school enrollment numbers[45], student-teacher ratios, and the average class size for schools[46]. For the network layer of workplaces, we used 2018 ACS 1-year estimates for county employment rates by age[47] and 2015 county estimates for workplace sizes[48].

In King County, a significant percent of COVID-19 cases and deaths have occurred to date within the LTCF population. However, most COVID models to date have not explicitly included LTCFs, which has been identified as a major limitation[49]. To capture the dynamics of transmission in this setting, the SynthPops model was extended to include the contact layer of those living and working in long term care facilities to reflect the initial outbreaks that occurred in these facilities[50].

Data on the demographics of LTCF residents for Washington state[51] was used to estimate the number of residents and their ages for Seattle area facilities. From these data, we estimate approximately 15,000 individuals aged 60 and older are residents of LTCFs within the Seattle area. Additional King County data on the number of residents per facility and resident-to-staff ratios were used to sample facility sizes and populate the facilities with both residents and staff members. There were an average of 123 residents per facility and 132 staff per facility, but there is wide variation in both total numbers of residents and staff and in the resident-to-staff ratios during any shift. For the purposes of this model, residents considered to be living in these facilities were not assigned any additional outside contacts (household, school, workplace, or community). Staff members were drawn from the labor force of the population under 60 years of age. With large facility sizes, we modeled close contacts in facilities by sampling for each resident and staff member a subset of 20 contacts, ensuring that each resident is in contact with at least one staff member.

Five synthetic populations of 225,000 modeled agents (representing 10% of King County's population) were generated using SynthPops, with dynamic scaling used to rescale this population to represent the full 2.25 million population of King County (see Section 2.6.2 of Kerr et al.[37]). These synthetic populations had roughly 500,000 household contacts, 1 million school contacts, 1.9 million workplace contacts, 4.5 million community contacts, and 31,000 LTCF contacts. To reflect the relative amount of time spent with each contact across different layers, averaged across a typical week, relative transmission weights per layer were set to be 100% for households (as a reference value), 50% for LTCFs, 20% for schools and workplaces, and 10% for community contacts. These values were chosen for consistency with both time-use surveys[52] and studies of infections with known contact types[36]. A subset of simulations were also run with 2.25 million agents to verify that results were consistent with and without rescaling.

**Calibration methodology.** We calibrated model parameters using Optuna 2.0.0, a Python-based optimization library[19], using the tree-structured Parzen estimator (TPE) sampler[53]. This sampler trains models of $p(\theta|y)$ and $p(y)$, where $\theta$ is a set of parameters and $y$ is a (scalar) output of a loss (objective) function, to find the region of the parameter space that minimizes $y$. We defined the loss function to be the sum of the absolute differences between observed data and the corresponding model predictions for seven different target outputs, namely: (a) cumulative diagnoses per day, (b) cumulative deaths per day, (c) 7-day rolling average diagnoses, (d) 7-day rolling average deaths, (e) total diagnoses by age (using 10-year age bins), (f) average test positivity rate by age, and (g) total deaths by age. To equalize the weight given to each point in each of these different data types, each data type was normalized by the maximum value in the data. In addition, cumulative vs. rolling average data were weighted in the ratio 4:1, which was found to most efficiently optimize the tradeoff between accurate fitting of long-term trends (driven by fits to cumulative data) and short-term trends (driven by fits to rolling average data). Deaths and diagnoses by age were given a relative weighting of 2, while test positivity by age was given a weight of 0.4 (since it was highly correlated with diagnoses by age, given that the total number of tests was fixed). These weights were chosen through an iterative process to determine algorithm convergence; final results are not sensitive to them, since the final simulations used for the analysis were good fits to all seven target outputs, and since they do not represent independent degrees of freedom (e.g., a good fit to rolling average diagnoses is necessarily at least a reasonable fit to cumulative diagnoses). Similarly, excluding a given target (e.g., diagnoses by age) did not always result in a significantly worse fit to that target, as long as at least one comparable target was included in the calibration (e.g., cumulative diagnoses).

We used 104,000 simulation runs during the calibration process to ensure broad exploration of parameter space. To determine parametric uncertainty, we used a cutoff value for the loss function of 30, which corresponded to no more than roughly 2% average relative error per point in the diagnoses and deaths time series, and roughly 10% average relative error for diagnoses, yield, and deaths age distributions; this cutoff was also roughly a factor of 2 larger than the single best-fitting simulations (which had total losses of 15.6 and 15.9 for calibrations with and without mobility data, respectively). Using this cutoff, the posterior distribution consisted of the 15,092 best-fitting parameter sets for the calibrations that used SafeGraph mobility data, and the 8821 best-fitting parameter sets for calibrations that did not. Median values and 95% confidence intervals for epidemic projections and parameter distributions (Fig. 1) were produced using these parameter sets (based on a uniform sample of 200 simulations). Detailed transmission characteristics (Fig. 2) were based on the single best fit with mobility data. For scenario analyses (Figs. 4 and 5), the top 10 best-fitting parameter sets for the calibrations that included mobility data were used.

Calibrating using four parameters was found to be sufficient to allow sufficient flexibility to capture observed epidemic trends, both with and without using mobility data as input. These parameters are shown in Table 1; uniform priors were used. Simulations were initialized with 300 seed infections, distributed at random throughout the population, on 27 January 2020. This initialization was chosen by calibrating the number of seed infections and overall transmission rate ($\beta$) to the subset of data prior to major policy or mobility changes (i.e., 27 February 2020), and for consistency with other estimates of the initial reproduction number in King County. In the calibration, larger numbers of seed infections were compensated for

**Table 1 Model parameters and calibrated values determined via fitting model outputs to King County data. CI, confidence interval.**

| Parameter | Primarily constrained by | Calibrated value with SafeGraph data: median (95% CI) | Calibrated value without SafeGraph data: median (95% CI) | Search interval |
|---|---|---|---|---|
| Probability of transmission per contact per day ($\beta$, %) | Initial rate of epidemic growth in observed diagnoses and deaths | 4.4 (4.2, 4.6) | 4.3 (4.2, 4.5) | [3.3, 4.8] |
| Relative reduction in transmission rate in work and community layers from 23 March 2020 onwards (%) | Numbers of diagnoses and deaths | 12.3 (2.9, 23.5) | 71.7 (65.0, 79.9) | [0, 90] |
| Relative reduction in transmission rate in LTCFs from 23 March 2020 onwards (%) | Age distribution of deaths and diagnoses; ratio of deaths to diagnoses; time trend of deaths | 86.1 (70.6, 94.2) | 87.1 (71.0, 94.3) | [60, 95] |
| Odds ratio of people with symptoms testing | Test positivity rate; number of diagnoses | 20.9 (15.2, 27.7) | 20.2 (14.8, 27.7) | [10, 60] |

by smaller transmission rates; 300 seed infections was the fewest that could be used (reflecting the highest baseline transmission rate) that provided a reasonable match to the data. We used SafeGraph data to determine the proportion of network edges in workplace and community layers that should be removed or restored over time based on observed changes in the mobility. Other model parameters were set to use Covasim defaults. The model was calibrated to data from 27 January until 9 June 2020. Scenarios began on 1 June 2020; we used the 9-day period of overlap to ensure consistency between calibrated and projected estimates of new infections, tests, diagnoses, and deaths

**Idealized test-trace-quarantine scenarios**. For the illustrative transmission trees shown in Fig. 3a–c, we used a hypothetical population of 100 people with a single seed infection simulated for 100 days. Population demographics were based on Seattle, Washington, USA, but contact networks were generated using the "hybrid" algorithm rather than SynthPops; this algorithm is described in Section 2.4.3 of Kerr et al.[37] Testing and tracing interventions began on day 20 of the simulation. The testing intervention used 15% daily probabilities of testing for people with symptoms; people without symptoms were not tested, and all people were tested upon entering quarantine. After consulting with Public Health Seattle & King County on estimated behavioral norms, contact tracing probabilities for the household, school, work, and community layers were set to 70%, 10%, 10%, and 0%, respectively (note that long-term care facilities are not included in these scenarios). People who were diagnosed and isolated were assumed to reduce their transmission rates by 70% for household contacts, and 90% for school, workplace, and community contacts. People who were contact traced and quarantined were assumed to reduce their transmission rates by 40% for household contacts, and 80% for school, workplace, and community contacts.

To explore the theoretical properties of test-trace-quarantine (Fig. 3d–f), we used a hypothetical population of 30,000 people with 100 seed infections simulated for 150 days. As above, a hybrid network was used. Simulations were run with 10 different random seeds, for three different transmission levels: medium transmission ($\beta = 4.2\%$ per household contact per day, consistent with estimated $\beta$ for Seattle, corresponding to $R_0 = 2.5$), low transmission ($\beta = 3.3\%$, $R_0 = 2.0$), and high transmission ($\beta = 5.1\%$, $R_0 = 3.1$).

Parameters for each of the three intervention scenarios (physical distancing, testing, and testing plus tracing) were chosen to bring $R_e \approx 1$ for the medium transmission scenario. These intervention parameters were held constant for the low and high transmission scenarios. The interventions that began on day 15 of the simulation for each of the three scenarios were:

1. *Physical distancing scenario*: 60% reduction in $\beta$, no testing or contact tracing;
2. *Testing scenario*: no reduction in $\beta$; daily probability of testing of 75% and 7.5% for people with and without symptoms, respectively, with no testing delay (test results returned same day); no contact tracing;
3. *Test-trace-quarantine scenario*: no reduction in $\beta$; daily probability of testing of 8%, 0.8%, and 75% for people with symptoms, without symptoms, and in quarantine, respectively, with no testing delay; tracing probability of 90% across all layers with no tracing delay.

While zero delays were used here, we also ran a sensitivity analysis with nonzero delays (1 day for testing and 1–2 days for contact tracing). Note that even with zero delays, there is a minimum one-day delay per step in the contact tracing process (since people who are placed into quarantine cannot test until the next timestep, i.e., the following day). In the model we assume that the duration of infectiousness is equivalent to the period during which a person would test positive. For idealized TTQ to succeed for high rates of transmission, the average delay for a single step of contact tracing must be less than the average serial interval, i.e., the average delay between a primary infection and a secondary infection. However, it is not necessary for the delays to be less than the shortest serial interval; if a secondary infection occurs prior to contact tracing, the cluster can still be contained as long as the average tracing delay is less than the average serial interval. An example of this is shown in Fig. 3c: person 82 is not diagnosed until after they have infected person 88 (day 82). However, person 88 is quarantined before they transmit further (day 93), and the cluster is contained.

**Realistic test-trace-quarantine scenarios**. We used the 10 parameter configurations with the best fit to the data over the period 27 January 2020 to 9 June 2020 as the basis for the test-trace-quarantine (TTQ) scenarios shown in Fig. 4a. Scenarios were also run with other sets of calibrations (including the top 100, and using the same goodness-of-fit threshold used for the distributions shown in Fig. 1e). These results did not differ qualitatively and only modest quantitative differences were observed; the top 10 calibrations were chosen to ensure the best fit to data while still capturing both parametric and stochastic uncertainty. To explore the relative importance of different intervention parameters, we ran a sweep of 50 points for each of the six parameters (described below), for each of the 10 parameter configurations, for a total of 3000 simulations. Each scenario began on 10 June 2020 and ended on 30 August 2020, which was chosen as the period prior to the potential reopening of schools. Scenarios began with an immediate return to 100% mobility in work and community (from a baseline value of 43% from the last reported SafeGraph data on 1 June 2020), as well as immediate implementation of

the testing and contact tracing interventions with the parameter values described below. Relative $\beta$ (compared to baseline), reflecting mask use and other non-pharmaceutical interventions, remained constant throughout the scenarios at its last calibrated value, which varied from 73% to 82%.

For each scenario, only one parameter at a time was varied. Note that the parameters interact nonlinearly; for example, the impact of the contact tracing delay depends on the amount of contact tracing. In addition, the impact of interventions is dependent on the epidemic dynamics: with very low mobility and hence low baseline transmission, the impact of interventions will be reduced. Thus, the baseline scenario was chosen to reflect (a) a situation where $R_e \approx 1$, which is the point most sensitive to small differences in intervention effectiveness; and (b) a balance between testing and contact tracing that is intended to reflect a realistic scale-up of both current programs. While other baseline points would be possible, this scenario aims to reflect a potentially achievable point by which Seattle-King County could maintain $R_e \approx 1$ with full reopening.

The six intervention parameters are defined as follows:

1. *Isolation/quarantine effectiveness*: The relative change in transmission following either diagnosis (isolation) or after being notified as a known contact (quarantine). While in practice (and for the assumptions used during the calibration period) the effectiveness would differ between isolation and quarantine (with isolation expected to have higher effectiveness), as well as between contact layers (with a greater reduction in workplace and community transmission compared to household), here we used a single weighted average value to ensure that the slope has meaningful units (i.e., infections averted per person fully isolated or quarantined). The default value chosen was 80% efficacy, which is a weighted average between workplace and community contacts (where isolation efficacy is likely to be higher, e.g., 90–95% effectiveness) and household contacts (where isolation efficacy is likely to be lower, e.g., 40–70% effectiveness). This parameter was varied from 0% (no impact of isolation/quarantine) to 100% (zero transmission during isolation/quarantine).
2. *Contact tracing probability*: The proportion of household, workplace, and LTCF contacts of a person who has been diagnosed who are reached by contact tracers. (The proportion of community contacts reached is assumed to be zero for this analysis; schools are closed for the scenario period so there are no school contacts to trace.) The default value chosen was 50%, which again reflects a weighted average between household and LTCF contact tracing (where probabilities well above 80% are achievable) and workplace contact tracing (where probabilities in Seattle-King County are currently low). This parameter was varied from 0% (no contacts traced) to 100% (all household and workplace contacts traced).
3. *Quarantine testing probability*: The probability that a known contact, once traced, will be tested for COVID-19. The default value used was 90%, regardless of symptoms. This parameter was varied from 0% (no testing of people in quarantine) to 100% (including uninfected, asymptomatic, and presymptomatic contacts). Upon testing negative, contacts were not released from quarantine, due to the possibility that they would become infected due to continued contact with the index case (as is often the case for household contacts), or in case they were exposed but had not yet started shedding at detectable levels.
4. *Routine testing probability*: The probability per person per day of receiving a test for COVID-19. The default values chosen were 16% per day for a person with active symptoms, and 0.16% for people who are uninfected or who do not have symptoms. These values correspond to an approximate doubling of the number of daily tests relative to 10 June 2020; the ratio of probabilities for people with and without symptoms was set to 100, which was chosen to be consistent with the observed testing yield in the data (approximately 1.5–2.5%). This parameter was varied from 0% (no routine testing) to 50% daily symptomatic testing and 0.5% daily non-symptomatic testing, corresponding to a roughly 4-fold increase in testing rates compared to 10 June 2020.
5. *Swab-to-result delay*: The average number of days between when a person receives a COVID-19 swab to when they are notified of their result. The default value chosen was 1 day, reflecting a slight improvement on practice in Seattle-King County as of June 10 (approximately 1.5 days). This parameter was varied from 0 days (immediate return of test results) to 7 days.
6. *Contact tracing delay*: The average number of days between when a person receives a positive result from a COVID-19 test and when their contacts are traced and notified. The default value chosen was 0 days, since most contacts (especially household contacts) are notified on the same day. This parameter was varied from 0 days (immediate notification of all contacts, although the swab-to-result delay is still present) to 7 days.

Because epidemic growth is an exponential process, the attack rate varied widely between scenarios, from less than 0.1% to nearly 50%. The attack rate had nonlinear dependence on all intervention parameters. Thus, the attack rate was log-transformed prior to fitting. The ordinary least squares method from the Python package *statsmodels* 0.12.2 was used for the fit. The uncertainty interval shown is the 95% confidence interval from the ordinary least squares fit. Because of the log transform, the slope of the line depends on the point of evaluation; in all cases, it

was evaluated at the default value for each parameter. Since the dependent variable in the regression is attack rate, the slope is also dependent on the period of integration (here, 91 days); a longer integration period, for example, would lead to a higher cumulative attack rate and thus larger slopes.

For the reopening sweeps (Fig. 4b), all parameters except for routine testing probability and contact tracing probability were fixed at the default values described above. We simulated eight different reopening levels (60% to 100%, in 5% increments), and show 60%, 80%, and 100% to represent low, medium, and high transmission scenarios. Each sweep consists of 12,000 simulations, with each simulation drawn from a uniform random distribution for (a) routine testing probability and (b) contact tracing probability, with each simulation also drawing from one of the 10 best calibrations as described above.

For reopening scenarios (Fig. 5), the status quo model was implemented using the baseline calibration to data until June 1, using input data on observed numbers of tests performed and contacts traced. In addition, mobility rates were increased to 80% on 1 June 2020, representing the lifting of the "Stay Home, Stay Healthy" measure; a subsequent $\beta$ reduction of 25% was applied on 1 July 2020, reflecting the statewide mask requirements that were mandated on 23 June and 7 July 2020, as described in the 13 August 2020 Situation Report.

**Reporting summary**. Further information on research design is available in the Nature Research Reporting Summary linked to this article.

## Data availability
All data used in this study are available via GitHub (https://github.com/amath-idm/controlling-covid19-ttq) and archived via Zenodo (https://doi.org/10.5281/zenodo.4699175). Data used in this study are also available from the King County Data Dashboard (https://www.kingcounty.gov/depts/health/covid-19/data.aspx), SafeGraph (https://safegraph.com), and the Seattle Coronavirus Assessment Network (https://scanpublichealth.org).

## Code availability
The Covasim model code is fully open-source and available on GitHub via https://covasim.org. SynthPops is also available on GitHub via https://synthpops.org. Analysis and plotting scripts to reproduce the results of this study are available via both GitHub (https://github.com/amath-idm/controlling-covid19-ttq) and Zenodo (https://doi.org/10.5281/zenodo.4699175)[41]. A webapp that uses this code to render interactive versions of the figures from this paper is available at https://ttq-app.covasim.org.

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

## Acknowledgements

The review of data and results was provided by Matthew Golden, Cathy Wasserman, and Ian Painter. Literature and code reviews were performed by Anna Palmer, Dominic Delport, Caroline S. Bennette, Bradley Wagner, Stewart Chang, and Edward Wenger. Additional contributors to the Covasim model and this study include: from GitHub, William Fitzgerald, Hamel Husain, Cory Gwin, Julian Nadeau, Rasmus Wriedt Larsen, Aditya Sharad, and Oege de Moor; from Microsoft, William Chen, Scott Ayers, and Rolf Harms; from the Institute for Disease Modeling, Mary Fisher, Amanda Izzo, Jennifer Schripsema, Dennis Chao, Christian Wiswell, Samuel Buxton, Christopher Lorton, Clinton Collins, Christopher Jones, Charles Eliot, Svetlana Titova, Dejan Lukacevic, Jeffrey Steinkraus, John Sheppard, Niket Thakkar, Roy Burstein, Robert Hart, Guillaume Chabot-Couture, Caitlin Bever, Helen Olsen, Greer Fowler, and Natalia Corona; from the Allen Institute, Natalia Orlova; from the Jet Propulsion Laboratory, Casey Handmer; from the QIMR Berghofer Medical Research Institute, Paula Sanz-Leon and James Roberts; from the Kirby Institute, Richard Gray; and from the Burnet Institute, Nick Scott and Sherrie Kelly. We also wish to thank the participants of the Covasim Users Group, including Julie Maher, Dean Sidelinger, and Erik Everson from the Oregon Health Authority; André Lin Ouédraogo from the Institute for Disease Modeling; and David P. Wilson from the Bill and Melinda Gates Foundation. Funding was provided by the Bill & Melinda Gates Foundation. Institutional support, including high-performance computing resources and library access, was provided by the Burnet Institute and the University of Sydney School of Physics.

## Author contributions

Covasim model development was led by C.K., R.S., R.A., and D.K., with additional support by G.H., K.R., P.S., R.N., J.C., L.G., and M.J. The SynthPops model was developed by D.M., with additional support by C.K., R.A., D.K., and L.G.. The health systems component was based on a model developed by B.H. Data were provided and curated by M.F. and J.D. Analyses were performed by C.K., D.M., R.S., K.R., G.H., R.N., P.S., and J.C. Supervision was provided by J.D., J.P.-G., M.F., and D.K.. The manuscript was written by C.K., D.M., R.S., R.N., and D.K. Manuscript review and editing was performed by K.R., J.C., J.P.-G., and M.F.

## Competing interests

The authors declare no competing interests.
