## [Peer Review File · Nature Communications]

Reviewers' Comments:

Reviewer #1:

Remarks to the Author:

The paper has improved a lot since I last read it (for another journal).

I have a comment still on the description of the idealised TTQ scenario (page 6) and related text in methods. You write that testing and tracing delays should be less than the period of infectiousness. Later you say that tracing delays should be smaller than the serial interval for tracing to be effective. Can you add some more information about your assumptions regarding latent and infectious periods and variable infectivity? Without this information these statements are hard to understand. I realise that you gave a reference to another paper, but I think this information is important enough to repeat here. Maybe also a figure that explains this relationship between delay and serial interval would be helpful. Also, this only works if tracing is performed over several steps, i.e. contacts of contacts are traced, etc. Correct? For backward tracing (what you call upstream), this delay would be too long, unless contacts of a source can then be traced very fast.

Reviewer #2:

Remarks to the Author:

In this paper, the authors explore the limitations of controlling COVID-19 using a test & trace strategy. Although the results are somewhat expected and have already been partially explored in the literature on COVID-19, I believe that both the methodology and the message that every aspect is important and that there is not a silver bullet solution are worthy.

I only have a few questions:

1) Is figure 1A a rolling average? I would expect a weekly pattern in the school layer and partially in the workplace layer. The effect in this specific scenario might not be too important since schools were closed soon, but to properly match simulation time to real-time this should be incorporated.

2) While reading the first part of the manuscript I got the impression that the model was fitted to just two curves (deaths and diagnoses) but then in the methods section, it is explained that 7 targets are used. This should be stated sooner. For instance, on lines 81-82 it is said "Covasim was able to accurately reproduce the detailed time trends of both diagnoses and deaths, including the age distribution of each". The model reproduces the trends and the age distribution because it is fitted to them, not saying so at this stage leads to thinking that it is a result of the model, but rather it is an input.

3) This is important because later on it is explained that "To reflect the relative amount of time spent with each contact across different layers, relative transmission weights per layer were set to be 100% for households, 50% for LTCFs, 20% for schools and workplaces, and 10% for community contacts".

More details are needed regarding these choices. For instance, in terms of time spent, transmissibility in LTCFs should be close to 100% since residents do not have contacts outside the facilities (albeit workers do). Similarly, if we assume a working shift of 8h, a 20% of transmissibility in that setting would imply that individuals spend 40h per day in households. Of course, the overall dynamics of the model will not change dramatically, but increasing 10% transmissibility in the community to 20% should have some effect on the numbers. I guess that the percentages reflect something else, not time spent, but this and the reasons behind this choice should be clearly stated.

Besides, and related to 1), if we were to increase the transmissibility in schools to 40%, or in LTCFs to 80%, the age distribution of cases should change accordingly, but it will not because the model is fitted to it. This is not a problem, but I do not think one can say that the model is able to accurately reproduce the input when it is being fitted to it.

4) I could not find any reference to the length of the isolation/quarantine, which varies a lot across countries and periods of time. I guess the one officially imposed in Seattle for that period is used, but this is an important aspect and should be explicitly defined. Furthermore, quarantined individuals who test negative are not released because they might be infected by the members of their households. I wonder if they are immediately released once the index case gets recovered. In general, the condition for releasing isolated/quarantined individuals should be better explained.

I could not find either if once a contact gets quarantined the whole household is also quarantined, but it looks like it given the previous policy.

Minor comments:

1) In the introduction it is said that the calibration uses data from January to June, but in the caption of figure 1, it says 1 February.

2) Why is a superspreading event defined as more than 20 infections? Besides, it seems that the definition (>20 infections per index case) refers to superspreading individuals, not to superspreading events.

3) There are several references to Section 2.X, which I was unable to find. I guess they come from a previous version of the paper.

4) Line 84: 27 January 27 -> 27 January

Response to reviewers: "Controlling COVID-19 via test-trace-quarantine"

We thank the reviewers for their helpful comments, all of which we agree reflect aspects of the manuscript that could be improved. We have modified the manuscript accordingly, and provide a detailed response to each comment below.

Please note that all code used for the manuscript is now available online at:

<https://github.com/amath-idm/controlling-covid19-ttq>

We will make this repository public if the manuscript is accepted for publication. In the mean time, it can be accessed by signing into GitHub using the following temporary account:

- Username: controlling-covid19-ttq
- Password: NCOMMS-21-06110_github

As part of this code release, we have also created a webapp that runs the code to produce interactive versions of each figure in the manuscript. This webapp, which is already public, is available here:

<https://ttq-app.covasim.org>

The code for this webapp is also included in the above repository. (Note: until the repository is made public, you will need to be logged into GitHub for the "View code" links to work.)

Reviewer 1

Comment	Response
The paper has improved a lot since I last read it (for another journal).	We are pleased to hear this, and thank the reviewer for their comments on the previous version as well.
I have a comment still on the description of the idealised TTQ scenario (page 6) and related text in methods. You write that testing and tracing delays should be less than the period of infectiousness. Later you say that tracing delays should be smaller than the serial interval for tracing to be effective. Can you add some more information about your assumptions regarding latent and infectious periods and variable infectivity? Without this information these statements are hard to understand. I realise that you gave a reference to another paper, but I think this information is important enough to repeat here. Maybe also a figure that explains this relationship	We recognize that this section is quite complex, and contains enough subtleties to be a manuscript of its own. Indeed, in the time since this manuscript was originally submitted (in July 2020), our team has been working on a separate manuscript on this topic (still in preparation), and several other authors have also published on it, including in high-profile journals, with similar findings to ours (e.g. Kojaku et al., Bradshaw et al., Endo et al.). We believe some of the confusion may have resulted from our imprecise use of the term "cluster". We have updated the text to refer to a "branch of the cluster", and added additional explanation of (a) the number of backwards steps that could be realistically taken, (b) the fact that it is still the forward process that determines the possibility of epidemic control (backward tracing makes the process more robust to imperfect diagnosis, quarantine, etc.). The reviewer is correct that if a "cluster" is interpreted to mean "all infections back to the original seed infection", then indeed (except for small clusters), backwards tracing is not guaranteed to find everyone in that cluster, since by the time the person corresponding to the seed infection is traced (indeed, even by the time

between delay and serial interval would be helpful. Also, this only works if tracing is performed over several steps, i.e. contacts of contacts are traced, etc. Correct? For backward tracing (what you call upstream), this delay would be too long, unless contacts of a source can then be traced very fast.	the first person in the branch is diagnosed), that person may no longer test positive. In light of this, we have significantly revised this section of the manuscript, including references to the three new papers mentioned above: "Before investigating TTQ in the Seattle context, we first consider how TTQ impacts SARS-CoV-2 transmission in a hypothetical population. Consider an idealized TTQ scenario, where all contacts are traced, all traced contacts are tested and enter into 14-day quarantine (regardless of test result), and combined testing and tracing delays are less than the duration of infectiousness (which is also assumed to be the time period when a person would test positive). In this idealized scenario, epidemic control can be achieved even for high values of R_0, regardless of the stage of the epidemic at which the intervention begins. This is because as a branch from a cluster of infections grows, the probability increases that someone from that branch will be diagnosed, and when this occurs, idealized contact tracing would identify that branch via a series of steps, including both backwards ("upstream") and forwards ("downstream") infections (26), hence removing that branch of the cluster from the infectious pool." "Since each traced contact who tests positive results in additional traced contacts, contact tracing can be thought of as an "infectious" process on the network. Specifically, if (a) the sum of the testing and tracing delays is less than the average serial interval of SARS-CoV-2; and if (b) the majority of secondary transmissions are successfully traced, diagnosed, and isolated, then the number of traced and diagnosed contacts will spread locally on the network faster than SARS-CoV-2 infections, extinguishing that branch. The number of backwards steps that can be taken is approximately the duration for which someone returns a positive test following infection divided by the sum of testing and tracing delays. Assuming the former is approximately 10–14 days and the latter is approximately 2–4 days, roughly 2–5 backward steps should be achievable, though in practice false negative tests would likely break the chain sooner. However, even with just forward tracing, epidemic control is still theoretically achievable. These results have been validated by several recent studies (26–28)."
--	---

Reviewer 2

Comment	Response
In this paper, the authors explore the limitations of controlling COVID-19 using a test & trace strategy.	We thank the reviewer for their positive comments. The reviewer is correct that weekday/weekend dynamics are not

Although the results are somewhat expected and have already been partially explored in the literature on COVID-19, I believe that both the methodology and the message that every aspect is important and that there is not a silver bullet solution are worthy. I only have a few questions: 1) Is figure 1A a rolling average? I would expect a weekly pattern in the school layer and partially in the workplace layer. The effect in this specific scenario might not be too important since schools were closed soon, but to properly match simulation time to real-time this should be incorporated.	explicitly included, albeit in several ways and for several reasons. Fig. 1A is a cumulative sum, which tends to obscure the weekly variability, although some differences can still be seen. (Note that the dots are plotted for every other day, since they would be too small to see otherwise.) For Fig. 1F, as the positions of the data points show, we only had weekly (not daily) mobility data available. For this reason (i.e. lack of mobility data), in the model (including Fig. 2A) we have not tried to differentiate weekday/weekend dynamics, and the values used represent weekly averages. The use of a weekly average value or day-specific value should have little impact on our results here, since the timescales we are looking at are much longer than a week. However, in a recent report that looks in detail at school reopening, we used a more finely detailed mobility model that included weekends (see Fig. 4 of that report). We have clarified this point in several places in the manuscript:  • In the Methods section we have added: "To reflect the relative amount of time spent with each contact across different layers, averaged across a typical week, relative transmission weights per layer were set to be 100% for households". • In the Results section: "calibrated the model both using reductions in the number of work and community contacts based on SafeGraph weekly mobility data (M, blue), and using no mobility data (N, red)". • In the limitations section of the Discussion: "First, we do not consider geographical clustering or day-of-the-week changes in mobility, so cannot model hotspots or outbreaks in specific areas or on specific days."
2) While reading the first part of the manuscript I got the impression that the model was fitted to just two curves (deaths and diagnoses) but then in the methods section, it is explained that 7 targets are used. This should be stated sooner. For instance, on lines 81-82 it is said "Covasim was able to accurately reproduce the detailed time trends of both diagnoses and deaths, including the age distribution of each". The model reproduces the trends and the age distribution because it is fitted to them, not saying so at this stage leads to thinking that it is a result of the model, but rather it is an input.	We have clarified this in the text: "We fit the Covasim model to age-stratified data on COVID-19 diagnosed cases and deaths in Seattle from January through June 2020". We note that although age-stratified data were used in the calibration process, in practice, only one of the four parameters (relative reduction in LTCF transmission) impacted the age distribution. Earlier calibrations which did not include the age data as explicit targets had fits to age distributions that were nearly as good; the inclusion of these additional targets was valuable primarily in that increasing the dimensionality of the objective function reduces the problem of local minima. We have added the following text to the methods: "Similarly, excluding a given target (e.g., diagnoses by age) did not always result in a significantly worse fit to that target, as long as at least one comparable target was included in the calibration (e.g., cumulative diagnoses)."

3) This is important because later on it is explained that "To reflect the relative amount of time spent with each contact across different layers, relative transmission weights per layer were set to be 100% for households, 50% for LTCFs, 20% for schools and workplaces, and 10% for community contacts". More details are needed regarding these choices. For instance, in terms of time spent, transmissibility in LTCFs should be close to 100% since residents do not have contacts outside the facilities (albeit workers do). Similarly, if we assume a working shift of 8h, a 20% of transmissibility in that setting would imply that individuals spend 40h per day in households. Of course, the overall dynamics of the model will not change dramatically, but increasing 10% transmissibility in the community to 20% should have some effect on the numbers. I guess that the percentages reflect something else, not time spent, but this and the reasons behind this choice should be clearly stated. Besides, and related to 1), if we were to increase the transmissibility in schools to 40%, or in LTCFs to 80%, the age distribution of cases should change accordingly, but it will not because the model is fitted to it. This is not a problem, but I do not think one can say that the model is able to accurately reproduce the input when it is being fitted to it.	We apologize for the omission of the explanation; it is provided in Section 2.3 of the methods manuscript that had been part of this paper when it was submitted previously. We have added the following text, based on the corresponding section of that manuscript: "These values were chosen for consistency with both time-use surveys (46) and studies of infections with known contact types (32)." Although these estimates were indeed based on time-use surveys, such surveys cannot be translated directly to a transmissibility value since the type of contact is different: for example, a 1 hour exposure to 1 of the 3 people in your household is likely to be much higher risk than a 1 hour exposure to 1 of the 20 people in your school, workplace, or LTCF. For example, given an average household size of 3 and an average workplace size of 12, we estimate the total daily risk of infection in both settings is roughly equal $((3-1)*100\% \approx (12-1)*20\% \approx 2)$. This is why we used contact type studies, in conjunction with time-use surveys, to estimate these proportions. However, we agree with the reviewer that these parameters are not very well constrained by data. One exception to that is the ratio of LTCF to household transmission; although this parameter was not explicitly calibrated, we did experiment with different values for it, and we found it was not possible to keep the age distribution of infections and deaths correct if the LTCF transmission rate per contact was equal to that of households, due to the order of magnitude larger number of daily contacts in LTCFs.
4) I could not find any reference to the length of the isolation/quarantine, which varies a lot across countries and periods of time. I guess the one officially imposed in Seattle for that period is used, but this is an important aspect and should be explicitly defined. Furthermore, quarantined	The reviewer is correct that we inadvertently omitted this. We have made the following amendments:  • In the Results, subsection "Idealized test-trace-quarantine results in self-limiting epidemic dynamics": "Consider an idealized TTQ scenario, where all contacts are traced, all traced contacts are tested and enter into 14-day quarantine" • In the Results, subsection "Realistic test-trace-quarantine scenarios allow high mobility": "(1) effectiveness of isolation and

individuals who test negative are not released because they might be infected by the members of their households. I wonder if they are immediately released once the index case gets recovered. In general, the condition for releasing isolated/quarantined individuals should be better explained. I could not find either if once a contact gets quarantined the whole household is also quarantined, but it looks like it given the previous policy.	quarantine (i.e., relative reduction in transmission during the 14-day isolation/quarantine period)"  In the Methods, subsection "Idealized test-trace-quarantine scenarios": "After consulting with Public Health Seattle King County on estimated behavioral norms, contact tracing probabilities for the household, school, work, and community layers were 70%, 10%, 10%, and 0%, respectively (note that long-term care facilities are not included in these scenarios). People who were diagnosed and isolated were assumed to reduce their transmission rates by 70% for household contacts, and 90% for school, workplace, and community contacts. People who were contact traced and quarantined were assumed to reduce their transmission rates by 40% for household contacts, and 80% for school, workplace, and community contacts." We hope this clarifies that a quarantined person has reduced risk of infecting others in the household, but the household itself is not quarantined. If the reviewer feels this point is still unclear in the manuscript, we are happy to add further clarification. We are exploring the impact of different quarantine lengths in a manuscript that is still in preparation.
Minor comments: 1) In the introduction it is said that the calibration uses data from January to June, but in the caption of figure 1, it says 1 February.	January 27 is correct; February 1 was from an earlier version. This has been fixed in the caption.
2) Why is a superspreading event defined as more than 20 infections? Besides, it seems that the definition (>20 infections per index case) refers to superspreading individuals, not to superspreading events.	This was intended as a rough clarification of what was meant by "superspreading", since no precise definition exists. However, we agree it was confusing and have removed it. Instead, we have updated Fig. 2 and the surrounding text to include calculations of how many infections are caused by individuals in different quantiles: e.g., the 25% of individuals who cause 80% of transmissions each infect 4 other people on average; the 10% of individuals who cause 50% of transmissions each infect 6.3 other people on average. We also flipped the axes of Fig. 2C and added additional annotations to make it clearer that it is the cumulative sum of Fig. 2B.
3) There are several references to Section 2.X, which I was unable to find. I guess they come from a previous version of the paper.	Thanks for catching this; the reviewer is correct. In the previous submission, the full Covasim methodology had been included as supplementary material, but it is now a separate manuscript (currently under review). The citations have been updated to refer to this manuscript.
4) Line 84: 27 January 27 -> 27 January	Fixed.

Reviewers' Comments:

Reviewer #1:

Remarks to the Author:

Thank you for your response and explanation, the idealised TTQ scenario has become more clear. However, I still doubt that up to five steps of backward tracing would be possible within the time window during which a case tests positive. There is also a latent period to consider, during which a case does not yet test positive, but which removes that person in time from the source, who does continue to transmit to others. If symptom onset occurs on average 5 days after infection, that is also the earliest time an index case would be found. Test sensitivity is very low in the first few days after infection and only increases shortly before symptom onset (see e.g. Kucirka et al 2020). So I agree that one step back is doable, but already two steps seems far fetched, because infection of the source occurred on average 10 days in the past.

The faster spreading of tracing as compared with the infection was also discussed in a perspective article that accompanied the Kojaku study (Mueller&Kretzschmar in Nature Physics).

Reviewer #2:

Remarks to the Author:

The authors have correctly addressed all my questions and I do not have any further comments.

Response to reviewers: "Controlling COVID-19 via test-trace-quarantine"

Reviewer 1

Comment	Response
Thank you for your response and explanation, the idealised TTQ scenario has become more clear. However, I still doubt that up to five steps of backward tracing would be possible within the time window during which a case tests positive. There is also a latent period to consider, during which a case does not yet test positive, but which removes that person in time from the source, who does continue to transmit to others. If symptom onset occurs on average 5 days after infection, that is also the earliest time an index case would be found. Test sensitivity is very low in the first few days after infection and only increases shortly before symptom onset (see e.g. Kucirka et al 2020). So I agree that one step back is doable, but already two steps seems far fetched, because infection of the source occurred on average 10 days in the past. The faster spreading of tracing as compared with the infection was also discussed in a perspective article that accompanied the Kojaku study (Mueller&Kretzschmar in Nature Physics).	We have added further clarification and a new citation on this point: "The theoretical maximum number of backwards steps that can be taken is approximately the duration for which someone returns a positive test following infection divided by the sum of testing and tracing delays. Assuming the former is approximately 10–14 days and the latter is approximately 2–4 days, roughly 2–5 backward steps are theoretically possible. In practice, false negative tests would likely break the chain sooner, although Japan, Vietnam, and Australia have successfully backwards-traced contacts for up to 14 days, in some cases by tracing secondary contacts before test results are returned (27). However, even with just forward tracing, epidemic control is still theoretically achievable. Several recent studies have produced similar findings (26, 28, 29)." We have also cited Müller & Kretzschmar as suggested. While rarely done in the US or UK, backwards tracing is standard practice in other countries. Vietnam traces contacts up to 5 steps removed and with exposure up to 14 days prior to the index case's diagnosis. Australia also has placed emphasis on finding "missing link" cases, with a recent example that gained media attention illustrating a 3-step backwards tracing. Thus, backwards tracing across multiple steps is not only theoretically achievable but has been put into practice by several countries – notably, countries that have done especially well at controlling their COVID-19 epidemics.

Reviewer 2

Comment	Response
The authors have correctly addressed all my questions and I do not have any further comments.	We are pleased to hear the comments have been addressed.